# Quantum Stream Cipher Based on Holevo–Yuen Theory: Part II

**DOI:** 10.3390/e26110983

**Published:** 2024-11-15

**Authors:** Osamu Hirota, Masaki Sohma

**Affiliations:** 1Quantum ICT Research Institute, Tamagawa University, 6-1-1, Tamagawa-gakuen, Machida, Tokyo 194-8610, Japan; sohma@eng.tamagawa.ac.jp; 2Research and Development Initiative, Chuo University, 1-13-27, Kasuga, Bunkyou-ku, Tokyo 112-8551, Japan

**Keywords:** quantum stream cipher, unicity distance theory, optical communication

## Abstract

This paper discusses the foundation of security theory for the Quantum stream cipher based on the Holevo–Yuen theory, which allows the use of “optical amplifiers”. This type of cipher is a technology that provides information-theoretic security (ITS) to optical data transmission by randomizing ultrafast optical communication signals with quantum noise. In general, the quantitative security of ITS is evaluated in terms of the unicity distance in Shannon theory. However, the quantum version requires modeling beyond the Shannon model of a random cipher to utilize the characteristics of the physical layer. Therefore, as the first step, one has to develop a generalized unicity distance theory and apply it to the evaluation of security. Although a complete theoretical formulation has not yet been established, this paper explains a primitive structure of a generalization of the Shannon random cipher and shows that the realization of this is the generalized quantum stream cipher. In addition, we present several implementation methods of generalized quantum stream ciphers and their security.

## 1. Introduction

Shannon has developed a historically groundbreaking theory for evaluating cryptographic functionality, utilizing his own entropy theory to analyze whether a cipher is decipherable [1]. Massey, on the other hand, offered a cryptographic direction that addresses real-world challenges while respecting Shannon’s concept [2]. The conditions that must be satisfied by ciphers for data to be useful in the real world are shown in Figure 1. That is, it is important that the ciphers used in practice satisfy the above in a balanced manner.

Massey suggested a concept to realize the above conditions as a technology, as follows:(1)A short symmetric key is set and it is extended by a pseudo-random number generator (PRNG).(2)The data are encrypted by the output of the PRNG.(3)Private randomization techniques are useful to attain information theoretic security.(4)Randomization for **plaintext** is a candidate for private randomization.

These are applicable for immunity against ciphertext-only attacks. A symmetric key cipher in modern mathematical cryptology requires at least a guarantee of computational security against known plaintext attacks. Thus, it was considered a difficult question whether or not an information-theoretic cipher can guarantee immunity against known plaintext attacks. To solve the above problem, H.P. Yuen proposed the basic concept of a new cipher, the so-called “generalized random cipher”, in collaboration with P. Kumar and O. Hirota; he disclosed a physical example of the generalized random cipher in a white paper in 2000, and the first experimental demonstration was reported in the QCMC in 2002 with the Kumar group [3]. The realization of the generalized random cipher is called the quantum stream cipher, at present. A more detailed name is the quantum noise randomized stream cipher operated by the Y-00 protocol. His fundamental idea consists of the following structure to satisfy Massey’s conditions of utility, as shown in Figure 1:(1)Use a high-powered laser with a coherent state as the light source and use optical amplifiers for long transmissions.(2)Use the mathematical cipher to diffuse the data into a set of *M*-ary coherent states that make up the transmission medium.(3)Adopt randomizations of **ciphertext** in order to effectively control the quantum noise of the high-powered laser.

What crucially distinguishes Yuen’s idea from Massey’s is that the randomization moves from plaintext to ciphertext to achieve information-theoretic security against known plaintext attacks (see Figure 2). To realize these configurations, Yuen proposed a scheme based on communication theory to differentiate between the physical signal detection capabilities of the legitimate receiver and the eavesdropper. This is called “advantage creation by secret key”. The way to perform this is to adopt a communication scheme such that the ciphertext received by the eavesdropper is hidden by quantum noise. Such a principle is called keyed communication in quantum noise (KCQ). This principle leads to a situation where the ciphertexts that can be received by the receivers for legitimate communicators and for eavesdroppers are different. Figure 3 shows the scheme of the conventional mathematical stream cipher. Figure 4 and Figure 5 show the scheme and the concept of the quantum stream cipher based on the Holevo–Yuen theory.

Applications of such a physical encryption technique appear to be limited, not applicable to general communication networks. But to diversify communication functions, a quantum stream cipher which utilizes physical phenomena is beginning to be considered for the ultrafast optical backbone network by several industries. However, this cryptographic technique has extremely complicated mechanisms, and some aspects have emerged that are difficult to communicate, even among experts in cryptography. The first paper in 2022 with the same title [4] has explained the position of the quantum stream cipher from a broad perspective as a cryptographic technique. In this second paper, we introduce the principles of security guarantees for quantum stream ciphers. Then, we describe the theoretical structure of security and explain its final target, as shown in Figure 6.

Since the main purpose of this paper is to explain how the random cipher based on the KCQ principle differs from the conventional Shannon random cipher, we first summarize Shannon’s concept in Section 2, which is based on information theory, and explain a generalized Shannon random cipher in this context as a conceptual model in Section 3. Then, we list the theoretical structure and performance of quantum stream ciphers such as Type-I, Type-II, and Type-III as concrete implementations of the conceptual model in the subsequent sections. In such models, we show that there exists a quantum stream cipher which is not broken, even if the secret key is stolen after communications.

## 2. Review of Shannon–Massey Theory of Cryptography

### 2.1. Historical View

It is well known that Shannon published in BSTJ, in 1949, his attempt to apply the entropy-based information theory to clarify a fundamental concept of cryptography. Many well-known researchers have explained its purpose [2,5,6,7], and the authors have nothing to add. However, in view of the development of cryptology after Shannon, it is worthwhile to explain the position of Shannon’s cryptology once again based on Massey’s concept [2].

Today’s practical cryptography has moved away from Shannon’s information-theoretic viewpoint to mathematical cryptography based on computer science, which has developed into the fundamental technology of the modern information society. One of the reasons for this is that the modern information infrastructure, of which the Internet is the main form of communication, is extremely large and cannot be handled only by Shannon’s cryptosystem. Mathematical cryptography based on computer science can handle large-scale systems and is absolutely indispensable for the ever-expanding communication infrastructure.

On the other hand, in recent years, ultrahigh-speed backbone communication networks have become necessary as a communication infrastructure to support huge systems represented by cloud computing, and thanks to the efforts of optical communication researchers, optical backbone communication lines are reaching the point of completion to support huge cloud computing systems. Against this backdrop, an environment has emerged in which information-theoretic cryptography can play an active role as a security technology for backbone communication networks.

Taking this opportunity, we believe it is worthwhile to analyze the Shannon–Massey scheme of cryptology again from various point of views. In the following sections, we describe our attempt.

### 2.2. Conventional Definition of Information-Theoretic Security for Symmetric Key Cipher

First, let us set a cipher mechanism to be considered. When a plaintext sequence is *X* and a secret key is *K*, the ciphertext is given by
(1)$$Y=F_{enc}\left(X,K\right)$$
where $F_{enc}$ is a encryption function. The secret key includes a running key sequence from PRNG with a short initial key or a true long random sequence.

We begin by stating a basic assumption of Shannon theory. In Shannon theory, when one constructs a cryptographic mechanism, there is an assumption that the ciphertext $Y^{A}$ is set by the legitimate sender and it can be correctly received by both legitimate receiver and eavesdropper. Namely,
(2)$$Y^{A}=Y^{B}=Y^{E}\equiv Y$$
where $Y^{B}$ and $Y^{E}$ are ciphertext sequences that can be received by the legitimate receiver and eavesdropper. Under this precondition, both the legitimate one and the eavesdropper can decrypt the correct ciphertext with the correct key sequence based on a secret key and PRNG. Such a mechanism can be expressed using the entropy theory developed by Shannon as follows:(3)$$H\left(X|K,Y^{B}\right)=H\left(X|K,Y^{E}\right)=0$$
Thus, the security of Shannon’s cryptology is defined as follows:

**Definition 1.** 
*An eavesdropper who does not know the secret key cannot decrypt the ciphertext $Y(=Y^{E}=Y^{B})$ with probability 1. This is equivalent to*

(4)
H(X|Y)≠0.

*This cryptosystem is said to be information-theoretically secure against a ciphertext-only attack (COA).*


On the other hand, the following theorem holds for cryptosystems with the conditions of Equation (Equation 2) [2,5,6].

**Theorem 1.** 
*When the ciphertext consists of plaintext and the key sequence, the ambiguity of the plaintext has the following limitations.*

(5)
H(X|Y)≤H(K)



This theorem is called the Shannon limit. In the following, we explain specific features of Shannon’s cipher under the conditions of Equation (Equation 2).

#### 2.2.1. Non-Random Cipher

Prepare a key sequence *K* to encrypt a plaintext sequence *X*. Assume that the ciphertext is composed of these two sequences. In this case, a non-random cipher is defined as follows:

**Definition 2.** 
*A cryptosystem whose constructed ciphertext satisfies the following properties is called a non-random cipher:*

(6)
$$H\left(Y^{E}|X,K\right)=H\left(Y^{B}|X,K\right)=0$$



A typical example of this type of cipher is an additive stream cipher. Let $X=x_{1},x_{2},x_{3},\ldots x_{n}$ and $K=k_{1},k_{2},k_{3},\ldots k_{n}$ be a plaintext and a key sequence, respectively. Then the ciphertext is as follows.
(7)$$Y^{A}=X⊕K=Y^{B}=Y^{E}$$

Here, Shannon gives the following definition of full confidentiality for the above cryptographic mechanism.

**Definition 3.** 
*A cryptosystem is fully confidential (or has perfect security in the sense of entropy) if it satisfies the following properties*

(8)
H(X|Y)=H(X)



Based on this definition, the conditions for achieving perfect security in the sense of entropy are as follows.

**Theorem 2.** 
*For a cipher system to have perfect security in the sense of entropy for any plaintext statistics, the key sequence length must be equal to or greater than the plaintext sequence length, i.e.,*

(9)
|X|≤|K|

*where the key sequence must be a sequence of true random numbers.*


The cryptographic mechanism that meets the above conditions is called the One Time Pad (or Vernum cipher). It is inefficient and leaves excesses in key management and known plaintext attacks [8,9].

#### 2.2.2. Shannon Random Cipher

In Shannon theory, we can consider an information-theoretically secure cipher even if it does not satisfy the full confidentiality condition. It is called a random cipher and is defined as follows:

**Definition 4.** 
*When a cryptosystem constructed under the conditions of Equations (2), (3), and (5) has the following properties*

(10)
H(Y|X,K)≠0

*it is called a random cipher in the sense of Shannon.*


In the case of a ciphertext-only attack, a cryptosystem with these properties can be realized by adopting a private randomization mechanism originating from Gauss, and its information-theoretic security is evaluated by the unicity distance theory. Its concrete structure is shown below [2,5,6].

### 2.3. Unicity Distance Theory

#### 2.3.1. Summary

Shannon and his successors have discussed cases where Theorem 2 does not hold but the encryption cannot be uniquely decrypted by a ciphertext-only attack. To simplify the discussion, we take a stream cipher encrypted by a pseudo-random number generator (PRNG) with a short secret key. In order to evaluate the degree of information-theoretic security of such a cipher system, the following unicity distance is defined.

**Definition 5.** 
*Let $Y_{n}=y_{1},y_{2},y_{3},\ldots ,y_{n}$ be a sequence of ciphertexts for an eavesdropper and $n_{0}$ be the minimum number of ciphertexts for which the ambiguity of the secret key is zero. Then, it is called the unicity distance of a ciphertext-only attack. That is,*

(11)
$$n_{0}:H\left(K|Y_{n_{0}}\right)=0$$



Here, we assume that the statistical structure of the plaintext is known, the entropy per symbol is H(p), and the entropy of the key sequence is H(K). Then, the number of key–plaintext pairs per each ciphertext is given by
(12)$$F=\frac{2^{H(K)}2^{nH(p)}}{2^{n}}=2^{H(K)-nH(p)-n}$$
Since F=1 in the above equation is equivalent to zero key ambiguity, the unicity distance is as follows [5,6]
(13)$$n_{0}=\frac{H(K)}{1-H(p)}$$
The above equation shows that the unicity distance depends on the statistical structure of the plaintext.

#### 2.3.2. Shannon Random Cipher and Its Unicity Distance

Homophonic substitution is an example of a method to increase the entropy per symbol of plaintext, independent of the key sequence. When this method is introduced, the ciphertext is not uniquely determined by the key–plaintext pair. In this case, Equation (Equation 10) holds. Therefore, this cipher system is a random cipher. In such a random cipher, the larger the entropy of the plaintext, the more difficult it becomes to estimate the key from the ciphertext. In other words, the unicity distance becomes larger. If H(X)=nH(p)=n, the unicity distance becomes infinite and the key cannot be uniquely determined even from an infinite number of ciphertexts. This is called an ideal cipher. However, if the eavesdropper obtains the correct key after obtaining the ciphertext, the correct plaintext can be decrypted since H(X|K,Y)=0 is a precondition. Thus, the Shannon limit holds.

### 2.4. Known Plaintext Attack Against Conventional Cipher

As introduced in the previous section, the cryptography theory of Shannon and his successors ends with a formulation for a ciphertext-only attack. In this section, we explain how Shannon’s theory holds when we assume a known plaintext attack (for simplicity, we assume an additive stream cipher).

**Definition 6.** 
*Let $Y_{n}=y_{1},y_{2},y_{3},\ldots ,y_{n}$ be the ciphertexts of length n of the eavesdropper, and assume that the plaintexts of the same length are known. The minimum number of ciphertexts for which the ambiguity with respect to the key is zero is defined as follows.*

(14)
$$n_{1}:H\left(K|Y_{n_{1}},X_{n_{1}}\right)=0$$

*The $n_{1}$ in the above equation is called the unicity distance of known plaintext attacks.*


In the following, we show some properties of the unicity distance for non-random ciphers and random ciphers in the Shannon theory.

#### 2.4.1. Non-Random Cipher

Since H(Y|K,X)=0 in a non-random cipher, the following holds if a ciphertext of length equal to the key length and the corresponding plaintext are known.
(15)$$H(K|Y_{\left|K\right|},X_{\left|K\right|})=0$$
where $Y_{\left|K\right|}$ and $X_{\left|K\right|}$ are the ciphertext equal to the key length and the corresponding known plaintext, respectively. From the above,
(16)$$n_{1}\leq \left|K\right|$$
This means that non-random ciphers can be decrypted in principle. Thus, the conventional symmetric key cryptography has only the above degree of information-theoretic security, and the rest relies on computational security.

#### 2.4.2. Shannon Random Cipher

For a random cipher in Shannon theory, one has H(X|K,Y)=0,H(X|Y)≤H(K) under the condition H(Y|K,X)≠0. From the theory of entropy, the following holds in general.
(17)$$H\left(X_{n}|Y_{n}\right)+H\left(K|X_{n},Y_{n}\right)=H\left(K|Y_{n}\right)+H\left(X_{n}|K,Y_{n}\right)$$
Assuming equality in the Shannon limit $H\left(X_{n}|Y_{n}\right)\leq H\left(K\right)$, there exists a finite *n*, namely $n_{1}$, for which $H(K|X_{n},Y_{n})=0$ from the above formula. Therefore, even if $n_{0}$ is infinite, a random cipher can be formally deciphered by a known plaintext attack. From the above, it is expected that conventional random ciphers are not very effective against known plaintext attacks because they depend on plaintexts, and if the structure of the plaintext is known, the randomization has little effect.

Thus, the conventional cryptographic techniques can provide information-theoretic security against a ciphertext-only attack, but these cannot provide information-theoretic security against known plaintext attacks.

## 3. Conceptual Generalized Random Cipher and Its Fundamental Properties

### 3.1. Conceptual Mechanism

Even with a cryptographic mechanism that uses a short secret key and PRNG, it is still possible to resist ciphertext-only attacks to a practical extent. In order to realize a cryptosystem with information-theoretic security for general attacks, it is necessary to develop a random cipher that goes beyond the existing concept of random ciphers. In this section, we provide an overview of the generalization of the Shannon random cipher as a conceptual mechanism.

Here, we define a generalized Shannon random cipher as a cryptosystem such that the ciphertext of the cryptosystem is hidden by ideal noise, and it has the following features.
(18)$$Y^{A}=Y^{B}\neq Y^{Eq}$$
where $Y^{A}$ and $Y^{B}$ are the ciphertexts of the legitimate communicator and $Y^{E_{q}}$ is the ciphertext received by the eavesdropper. In other words, this is achieved by creating a situation where the ciphertext received by the legitimate receiver and the ciphertext that can be received by the eavesdropper are different. If we describe this in a sequence, it is as follows:(19)$$\begin{array}{ccc} & & Y_{n}^{B}=y_{1}^{A},y_{2}^{A},y_{3}^{A},\ldots ,y_{n}^{A} \end{array}$$(20)$$\begin{array}{ccc} & & Y_{n}^{Eq}=y_{1}^{E_{q}},y_{2}^{E_{q}},y_{3}^{E_{q}},\ldots ,y_{n}^{E_{q}}=y_{1}^{A}⊕q_{1},y_{2}^{A}⊕q_{2},y_{3}^{A}⊕q_{3},\ldots ,y_{n}^{A}⊕q_{n} \end{array}$$
where $q_{1},q_{2},q_{3},\ldots$ denote errors due to true noise. If this situation could be realized, then the following conceptual relations would be held.
(21)$$\begin{array}{ccc} & & H(Y^{B}|K,X)=0, \end{array}$$
(22)$$\begin{array}{ccc} & & H(Y^{Eq}|K,X)\neq 0 \end{array}$$

Such a random cipher is a completely different form of random cipher than the conventional Shannon random cipher. Why this is possible is explained in a later section.

### 3.2. Generalized Unicity Distance

First, if the ciphertext received by a legitimate receiver and the ciphertext that can be received by an eavesdropper are different, the following situation is possible in the practical sense.
(23)$$\begin{array}{ccc} & & H(X_{n}|K,Y_{n}^{B})=0 \end{array}$$
(24)$$\begin{array}{ccc} & & H(X_{n}|K,Y_{n}^{Eq})\neq 0 \end{array}$$
In other words, the legitimate receiver can obtain the correct plaintext with the correct key, but because the ciphertext of the eavesdropper has errors, the possibility appears such that the correct plaintext is not obtained even with the correct key. We need a generalization of the unicity distance to evaluate the security of the system in such cases. The unicity distances for such a cryptographic mechanism are described below [10,11,12].

#### 3.2.1. Ciphertext-Only Attack

The unicity distance of a ciphertext-only attack is described for the eavesdropper’s ciphertext as follows:

**Definition 7.** 
*Let $n_{0}^{Q}$ be the minimum length of the ciphertext that has zero key ambiguity for the eavesdropper’s ciphertext. Then, it is given by*

(25)
$$n_{0}^{Q}:H\left(K|Y_{n_{0}^{Q}}^{Eq}\right)=0$$

*$n_{0}^{Q}$ is called the unicity distance of the ciphertext-only attack for a generalized random cipher.*


Unlike the conventional type, the above equation does not depend on the statistical structure of the plaintext, but on the randomness of the ciphertext that can be obtained by the eavesdropper. The error sequence in Equation (Equation 20) should be completely random numbers or its equivalent.

To achieve this, it is convenient to use the quantum noise effects of light. As an example, if the signal system consists of non-orthogonal quantum states when the eavesdropper receives the signal, the theory is that an ideal random number error effect will appear in the received signal. This is due to quantum irregularities (Born effect) when quantum superpositions are collapsed by measurement. Detailed discussions will be given in the subsequent sections.

#### 3.2.2. Known Plaintext Attack

The conventional random ciphers can achieve a large unicity distance for a ciphertext-only attack, but the technology to achieve this is very complicated. Furthermore, it is difficult to guarantee information-theoretic security more than a key length in the known plaintext attack.

Here, we consider a known plaintext attack on a generalized random cipher. First, the information-theoretic security evaluation for the known plaintext attack is given as follows.

**Definition 8.** 
*The unicity distance of known plaintext attacks for generalized random ciphers is defined as follows:*

(26)
$$n_{1}^{Q}:H\left(K|X_{n_{1}^{Q}},Y_{n_{1}^{Q}}^{Eq}\right)=0$$



In the case of the generalized random cipher, since it does not depend on the structure of the plaintext, the known plaintexts do not have much effect on the unicity distance. In other words, the following can be expected.
(27)$$\left|K\right|\ll n_{1}^{Q}\leq 2^{\left|K\right|}$$
Later sections will discuss specific examples of this.

#### 3.2.3. Secret Key Leakage Attack in COA

In conventional symmetric-key ciphers, an eavesdropper can retain the correct ciphertext, and if she can obtain the secret key after the communication, the correct plaintext can be obtained. This is due to H(X|K,Y)=0. However, in generalized random ciphers, a possibility of Equation (Equation 24) arises because the eavesdropper’s ciphertext will be inaccurate. Therefore, we can define the following evaluation function.

**Definition 9.** 
*When the secret key (initial key for PRNG) is stolen after communication, the minimum length of ciphertext needed by the eavesdropper to obtain the correct plaintext is as follows:*

(28)
$$n_{2}^{Q}:H\left(X_{n_{2}^{Q}}|K,Y_{n_{2}^{Q}}^{E_{q}}\right)=0$$

*The above is called the unicity distance of the secret key leakage attack.*


## 4. Concrete Evaluation Method of Generalized Shannon Random Cipher

In the above section, we defined several unicity distances for several attacks. The main purpose of a generalized Shannon random cipher is to improve security against known plaintext attacks (KPAs). In order to perform a quantitative evaluation of security, it is necessary to develop a method for its calculation. We discuss a method in the following.

In the case of KPA for the conventional cipher, an eavesdropper can obtain the correct running key sequence which corresponds to the output sequence from the PRNG.

The purpose of the eavesdropper is to estimate the secret key of the PRNG from the running key sequence. Assume that the PRNG consists of a linear shift register (LFSR) with a secret key (e.g., 256 bits) and a nonlinear filter. In general, one can adopt information theoretic analysis in an immunity evaluation in the conventional stream ciphers. The technique is called a fast correlation attack.

The efficiency of key estimation is evaluated by considering the following model. Suppose that the LFSR output is regarded as a linear code word of (n,|K|) and the nonlinear filter section is modeled as a noisy communication channel with an error rate of ϵ which depends on the structure of the nonlinear part. So, the model corresponds to a decoding problem for a linear code word of length *n* and information bit *K*. The feature of this method is that the computational security of the nonlinear filter section is analyzed by an information-theoretic model. Such a theory was developed by Siegenthaler [13], Chepyzhov-Smeets [14], and others, from which the following theorem was derived [13].

**Theorem 3.** 
*When the length n of the code word, which corresponds to the length of the output sequence obtained by the eavesdropper, is satisfied as follows:*

(29)
$$n>N=\frac{\left|K\right|}{C(\epsilon )}$$

*then, the probability of a key being correct is greater than 1/2 if the total search complexity of the decryption algorithm is*

(30)
$$O(2^{\left|K\right|}\times \frac{\left|K\right|}{C(\epsilon )})$$

*where C(ϵ) is the maximum mutual information of the communication channel model with error ϵ.*


Although this body of theory analyzes computational problems in the context of information theory, it was pointed out by Yuen et al. that these theories are rather appropriate for the ITS analysis of generalized random ciphers [10,11]. Here, we apply this theory to a detailed characterization of generalized random ciphers.

The generalized random cipher consists of a PRNG with a short key and real noise for randomization of the ciphertext or running key sequences. The nonlinear filter part of the running key sequence of the generalized random cipher does not make sense from an information-theoretic point of view. Therefore, the model is replaced as follows.

A sequence of LFSR with an initial key is the channel input, and this sequence is perturbed by true noise and a channel model by the nonlinear part. But, the main issue is a perturbation by real noise. Then, the procedure of the eavesdropper is to use decoding theory for the code word (LFSR) based on the sequence with errors due to the true noise.

Conventional theories that use communication channel models for computational problems lose the rigor of applying code theory because the noise model is not an independent process, but in a generalized random cipher, noise is a true independent process, so the above idea is more valid. From the above, the aforementioned theorem can be read into the problem of information-theoretic cryptography.

As a result, if we consider the meaning of the unicity distance, we can set up the following relation.
(31)$$n_{1}^{Q}\geq \frac{\left|K\right|}{C(\epsilon_{q})}$$

$C(\epsilon_{q})$ is the maximum mutual information of the measurement channel where the eavesdropper receives the running key sequence. $\epsilon_{q}$ is the error probability due to true noise.

## 5. Protocol and Structure of Standard Quantum Stream Cipher

The challenge is how to show that the conceptually presented generalized random ciphers are real. The Y-00 protocol was proposed to solve this problem. The Y-00 protocol is a technique that combines cryptographically secure pseudo-random numbers and quantum noise effects, and is characterized by the fact that it can be equivalently regarded as a function to hide the ciphertext of mathematical ciphers with quantum noise. In this section, we show a simple explanation of the original scheme of a quantum stream cipher based on the Y-00 protocol [10,11], which is called a basic quantum stream cipher or standard quantum stream cipher.

In order to guarantee ultrahigh-speed and long-distance transmission, it is necessary to adopt optical signals with high energy, not single photon or entanglement light. However, in general, high-power signals have little quantum effect and do not adequately hide the ciphertext. Therefore, we introduce a mechanism in which the receiver Bob (the legitimate receiver) can ignore the quantum effect, such as quantum noise, and the receiver Eve (the eavesdropper) cannot avoid the quantum effect, even though the light is strong. This scheme is called “Advantage creation by secret key”. It is designed as follows:
(a)Let us consider two optical signals with quantum coherent states sending 0 and 1 data. Let us denote the two coherent state signals as follows: |α> and |−α>, with |α|≫1, which is an amplitude of laser light. A pair of two coherent states is called the communication basis Ba(g) for transmitting the “Binary Signal Data” as plaintext, where g=1,2,3,…,M. It means to prepare a set *M* of different pairs Ba(g) of two coherent states with different complex amplitudes, as follows:
(32)$$\begin{array}{ccc} & & Ba\left(1\right)=\{|\alpha e^{i\theta_{1}}>,|-\alpha e^{i\theta_{1}}>\},\\ & & Ba\left(2\right)=\{|\alpha e^{i\theta_{2}}>,|-\alpha e^{i\theta_{2}}>\},\\ & & \vdots \\ & & Ba\left(M\right)=\{|\alpha e^{i\theta_{M}}>,|-\alpha e^{i\theta_{M}}>\} \end{array}$$The selection can be realized by a unitary transformation controlled by the running key. In the standard quantum stream cipher, the data information x=0,1 are assigned regularly to one of the signals of each basis. Thus, in general there is a correlation between the plaintext and running key as in the conventional cipher.(b)Alice and Bob share the same PRNG with the same secret key (for example 256 bits) as the conventional stream cipher. A sequence is generated by the output of PRNG. This is called a “running key” sequence in a stream cipher. Alice’s transmitter selects a communication basis in Equation (Equation 32) following the running key of *M* values, and then one of the binary data is transmitted by using the selected communication basis. Thus, a sequence of 2M-ary optical signals with coherent states of different amplitudes or phases is transmitted. It is a quantum ciphertext that must be converted into an electrical signal by quantum measurement for cryptanalysis.(c)Alice and Bob share the same running key, so Bob can know which basis was selected. That is, he can receive the optical signal as the binary signal after an inverse unitary transformation. So, the quantum signals for Bob are as follows:
(33)$$\rho_{0}^{B}=|\alpha ><\alpha |,\;\rho_{1}^{B}=\left|-\alpha ><-\alpha \right|$$This is independent of the communication basis.But Eve has no information on the running key, because she does not know the secret key for PRNG. So Eve has to use a receiver for 2M-valued signals, and has to discriminate 2M-ary phase shift keying (PSK) signals. In the case of phase shift keying, the set of quantum states is described by
(34)$$\rho_{m}^{E}=\left|\alpha e^{i\theta_{m}}><\alpha e^{i\theta_{m}}\right|,\;m=1,2,3,\ldots ,2M$$
where *m* is controlled by the binary data and *M*-ary running key. Thus the error performance of Bob is given by binary quantum detection, and the error performance of Eve is formulated by the 2M-ary quantum detection for cryptanalysis.(d)The Y-00 protocol requires a signal constellation such that the binary detection is error free, but the 2M-ary signal detection suffers from the quantum noise effect based on the Helstrom–Holevo–Yuen principle [15,16,17,18]. That is, a non-orthogonal quantum state signal cannot be discriminated without error. The above structures satisfy this condition, because a binary detection of two coherent states is regarded as nearly orthogonal for |α|≫1, but the 2M-ary detection for the complex amplitude $\alpha_{m},m=1,2,3,\ldots ,2M$ is regarded as a non-orthogonal quantum state system. The concrete signal constellation based on phase shift keying (PSK) is given in [10].(e)Consequently, Bob can obtain directly a data bit sequence without serious error. However, Eve can only obtain a multi-level signal sequence which corresponds to ciphertext as the measurement result of quantum ciphertext. This electrical sequence of the ciphertext has errors. So Eve has to recover the data sequence or secret key of the PRNG from this sequence with errors.

Thus a quantum stream cipher based on the Y-00 protocol is a candidate for the generalized random cipher in which Eve cannot obtain the correct ciphertext (see Figure 7). Moreover, this scheme has technical advantages in real world applications. That is, the noise for randomization is only generated in the measurement process, and it does not disturb the bandwidth of the channel or data speed of the legitimate communicator. Thus, it is applicable to the conventional optical communication systems for ultrafast data transmission.

## 6. Quantum Communication Theory for Cryptanalysis

The security of the cipher relies on errors in the ciphertext that an eavesdropper can obtain. Therefore, quantum communication theory plays an essential role in analyzing errors in the ciphertext received by the eavesdropper. In the following, we will denote a formulation of the error analysis of the ciphertexts of legitimate receivers and eavesdroppers based on the quantum communication theory.

### 6.1. Fundamental Formulae

The quantum communication theory was initiated in the 1970s and 1980s by pioneers such as C. Helstrom, R. Kennedy, A. Holevo, V. Belavkin, H.P. Yuen, S. Personic, and V.W.Chan, and its whole formulation was integrated by Helstrom [15] and Holevo [16]. Specifically, Holevo [17] and Yuen [18] clarified the optimum conditions of the quantum Bayes detection rule for multi-level signals independently, and Hirota–Ikehara formulated the quantum minimax detection rule with the admissibility and completeness [19] that corresponds to the quantum version of the Wald–Middleton decision theory [20,21]. An introduction is available in [22]. Let us describe the formulation of quantum detection theory of the core of quantum Shannon theory. When the 2M-ary coherent state signal is received at each slot, the optimizing variable of the quantum measurement channel is described by a compact set of the positive operator-valued measure (POVM): $\Pi_{m}$, m=1,2,3,…,2M. Then, these operations are interpreted as the projector acting on the quantum state of each slot, and these provide error or detection probabilities as follows:(35)$$\begin{array}{ccc} P(\alpha_{l}|\alpha_{m}) & = & Tr\rho_{m}\Pi_{l},\;m,l=1,2,3,\ldots ,2M\\ \rho_{m} & = & |\alpha_{m}><\alpha_{m}|,\;\sum_{l}\Pi_{l}=I,\;\Pi_{l}\geq 0\;\forall l \end{array}$$
The appearance of quantum effects in the reception process of signals is characterized by the above formula. The quantum Bayes rule is formulated as follows:(36)$${\bar{P}}_{e}=\min_{\left\{\Pi \right\}}\left\{1-\sum_{m=1}^{2M}\xi_{m}Tr\rho_{m}\Pi_{m}\right\}$$
where a priori probability is $\left(\xi_{m}>0,\;\forall m\right)$ for the admissibility in the decision theory. The necessary and sufficient conditions are given as follows [17,18]:

**Theorem 4.** 
*{Holevo,Yuen}:*

(37)
$$\begin{array}{ccc} & & \Pi_{m}\left[\xi_{m}\rho_{m}-\xi_{l}\rho_{l}\right]\Pi_{l}=0,\;\forall l,m\\ & & \gamma -\xi_{l}\rho_{l}\geq 0,\;\forall l\\ & & \gamma =\sum_{l}\xi_{l}\rho_{l}\Pi_{l} \end{array}$$



On the other hand, the quantum minimax rule is formulated as follows [19]:(38)$${\bar{P}}_{e}=\max_{\left\{\xi \right\}}\min_{\left\{\Pi \right\}}\left\{1-\sum_{m=1}^{2M}\xi_{m}Tr\rho_{m}\Pi_{m}\right\}$$
The necessary and sufficient conditions are given as follows [19]:

**Theorem 5.** 
*{Hirota·Ikehara}:*

(39)
$$\begin{array}{ccc} & & Tr\Pi_{l}\rho_{l}=Tr\Pi_{m}\rho_{m},\;\forall l,m\\ & & \Pi_{m}\left[\xi_{m}\rho_{m}-\xi_{l}\rho_{l}\right]\Pi_{l}=0,\;\forall l,m\\ & & \gamma -\xi_{l}\rho_{l}\geq 0,\;\forall l\\ & & \gamma =\sum_{l}\xi_{l}\rho_{l}\Pi_{l} \end{array}$$



In general, it is very difficult to find the solutions of the above two quantum detection rules. However, in the standard quantum stream cipher system, quantum state signals have a property of the covariant as defined below.

**Definition 10.** *Let G be a group with an operation* ∘*. The set of quantum state signals is called the group covariant if there exist unitary operators $U_{k}\left(k\in G\right)$ such that*
(40)$$U_{k}\left|\psi_{m}>=\right|\psi_{k\circ m}>,\forall m,k\in G$$
*It characterizes the quantum states $\left\{|\psi_{m}>,m\in G\right\}$.*

The general properties of the quantum Bayes rule for the covariant case of multi parameters are given by Ban [23]. One of the results for coherent state signals is as follows:

**Theorem 6.** 
*If the signal set $\left\{|\alpha_{m}>\right\}$ is a covariant, the optimum POVM is given by using Gram operator H as follows:*

(41)
$$\Pi_{l}=|\mu_{l}><\mu_{l}\left|,\;\right|\mu_{l}>=H^{-1/2}|\alpha_{l}>,\;H=\sum_{m=1}^{M}\left|\alpha_{m}><\alpha_{m}\right|$$

*and the optimum quantum Bayes solution is*

(42)
$${\bar{P}}_{e}=1-|<\alpha_{1}|H^{-1/2}{\left|\alpha_{1}>\right|}^{2}$$

*where $|\alpha_{1}>$ is the base state.*


The error probability for Eve for 2M covariant signals can be given as follows [15,24,25]:(43)$$\begin{array}{ccc} {\bar{P}}_{e}^{E} & = & 1-\frac{1}{{\left(2M\right)}^{2}}{\left(\sum_{m=1}^{2M}{\sqrt{\lambda }}_{m}\right)}^{2}\\ \lambda_{m} & = & \sum_{k=1}^{2M}<\alpha_{1}|\alpha_{k}>u^{-(k-1)m} \end{array}$$
where u=exp[πi/M]. In addition, Osaki showed that the worst a priori probability in the quantum minimax rule for the covariant signals becomes the uniform distribution and the minimax solution is also given by Equations (41) and (42) [26].

### 6.2. Advantage Creation by Differentiation of Quantum Detection Performance by Secret Key

Let us apply the above formulae to cryptanalysis. Bob can control the unitary transformation to convert back to binary optical signals by using the running key from the pseudo-random number for the 2M optical signals. Then, the quantum detection model becomes the binary quantum states of $\left\{\rho_{0}^{B},\rho_{1}^{B}\right\}$: Equation (Equation 33), independent of the communication basis. The average error probability is given by the Helstrom formula, as follows [15]:(44)$${\bar{P}}_{e}^{B}=\min_{\left\{\Pi \right\}}\left\{1-\sum_{m=0}^{1}\xi_{m}Tr\rho_{m}^{B}\Pi_{m}\right\}=\frac{1}{2}\left[1-\sqrt{1-4\xi \left(1-\xi \right)Tr\left(\rho_{0}^{B}\rho_{1}^{B}\right)}\right]\ll \frac{1}{2}$$

On the other hand, “in order for Eve to perform the cryptanalysis”, she has to obtain the information of the running key sequence by her quantum measurement to the 2M-ary quantum ciphertext. The first step in the procedure leading to an attack is to receive a signal flowing through the real communication channel. The average minimum error probability for the adopted quantum state signal scheme (or equivalently, the maximum detection probability) can be given by the following formulae: Equations (38), (39), (41), and (43). For M≫1, it becomes
(45)$${\bar{P}}_{e}^{E}=\max_{\left\{\xi \right\}}\min_{\left\{\Pi \right\}}\left\{1-\sum_{m=1}^{2M}\xi_{m}Tr\rho_{m}\Pi_{m}\right\}∼1$$
Thus, these formulae provide the theoretical accuracy of the ciphertext that the eavesdropper can obtain.

If Eve were to attempt to decode the binary data directly, she would adopt the binary quantum optimal measurement for the following mixed quantum states.
(46)$$\begin{array}{ccc} \rho_{0}^{E} & = & \frac{1}{M}\sum_{m=1}^{M}\left|\alpha_{\left(m=even\right)}><\alpha_{\left(m=even\right)}\right|\\ \rho_{1}^{E} & = & \frac{1}{M}\sum_{m=1}^{M}\left|\alpha_{\left(m=odd\right)}><\alpha_{\left(m=odd\right)}\right| \end{array}$$
This structure of mixed states is called doubly symmetric mixed state, and Kato gives the quantum Bayes (also minimax) solution for such as mixed states of the coherent state [27]. Then, the average error probability for binary data is given as follows:(47)$${\bar{P}}_{e}^{E}=\max_{\left\{\xi \right\}}\min_{\left\{\Pi \right\}}\left\{1-\frac{1}{2}\sum_{l=0}^{1}Tr\rho_{l}^{E}\Pi_{l}\right\}∼\frac{1}{2},M\gg 1$$

The difference between Equation (Equation 44) vs. Equation (Equation 45) and Equation (Equation 44) vs. Equation (Equation 47) is called the advantage creation by the secret key.

### 6.3. Physical Processes for Cryptanalysis Against PRNG

A sequence of length *n* of quantum ciphertext from a transmitter of a standard quantum stream cipher is described as follows:(48)$$\rho^{E}\left(x_{1},k_{1}^{R}\right)⊗\rho^{E}\left(x_{2},k_{2}^{R}\right)⊗\cdots ⊗\rho^{E}\left(x_{n},k_{n}^{R}\right)$$
x∈X are binary data (plaintext) and $k^{R}\in K^{R}$ is the running key of *M* values from PRNG. Let us denote the physical attack process for cryptanalysis in the following.

#### 6.3.1. Individual Quantum Measurement-Collective Procedure

Let us assume that Eve adopts a quantum optimum measurement $\left\{\Pi_{i}\right\}$ for each slot. The observed sequence corresponds to a sequence of the decision output for the 2M-valued signal at each slot. The randomness of signals is represented by Equation (Equation 38), and its randomness automatically provides a fully independent true random noise. The target for Eve is data (binary plaintext) or the secret key of PRNG, and she has to estimate them from the sequence with errors based on the security analysis like in the correlation attack or other attack. Such a procedure is called an “individual quantum measurement-collective attack”.

#### 6.3.2. Collective Quantum Measurement-Collective Procedure

On the other hand, Eve can adopt a collective quantum measurement. It is a quantum measurement such that one treats some slots of the coherent state sequence of 2M-ary as one block quantum state. Here, Eve has to construct a quantum entanglement measurement system of ${\left(2M\right)}^{{\left|N\right|}_{Bl}}$ signals, where ${\left|N\right|}_{Bl}$ is the length of the block. Based on such a measured sequence, she analyzes several attacks against the sequence. This scheme is called a “collective quantum measurement-collective attack”. When ${\left|N\right|}_{Bl}$ is a large number, it seems that this physical implementation is impossible. After such physical manipulations, the eavesdropper would be forced to make some crypt analytic attempts.

## 7. Cryptological Attack for Quantum Stream Cipher and Its Performance

### 7.1. The Main Attack Schemes of Symmetric Key Ciphers with PRNG

The PRNG used in this cipher is guaranteed to be computationally secure. If the eavesdropper’s error in the quantum stream cipher is zero, then the security is consistent with the security of the PRNG itself. Therefore, the security analysis of the quantum stream cipher is based on the error in the ciphertext received by the eavesdropper, which is an investigation of how the cryptanalysis of the mathematical cipher is invalidated. This is because the purpose of this cipher is to make the mathematical analysis and exhaustive search for symmetric key ciphers impossible. Thus, it is sufficient to mention exhaustive search, correlation attacks, and key compromise as attack methods for the discussion of information-theoretic security against symmetric key ciphers.

Here, let us describe briefly the concept of cryptological attacks against the standard and some generalized quantum stream ciphers. At first, we denote a notion of the quantum noise effect in the Y-00 protocol. Since the receiver Bob adopts a binary detection scheme by the communication basis synchronized with the same PRNG with the same secret key, there is no mismatch in the communication basis. Then, it outputs a binary signal as data without serious error, because the signal power is strong and the signal distance of the two signals in the basis is large (See Figure 4). However, Eve’s received signals are 2M-ary and contain errors, because the signal distances between several signals are small. The extent of the error is called the noise masking region (See Figure 8).

**Definition 11.** *Let “ *Γ *” be the number of signals masked by quantum noise in several measurement processes. Then, 1/Γ is the correct probability of signals in the wedge approximation.*

#### 7.1.1. Legitimate Receiver Simulation Attack

Let us consider the KPA based on the exhaustive search. This attack clarifies the difference between the standard symmetric key cipher and quantum stream ciphers. In the standard quantum stream cipher based on the Y-00 protocol, the data (plaintext) of the signal of each basis are deterministically set such that data 0 and 1 are regularly mapped to neighboring signals composed of a communication basis [10]. That is, the data information is mapped to 0,1,0,1,… clockwise of the phase signal in the phase space. This means that the standard quantum stream ciphers have a correlation between basis and data.

As a result, each signal of 2M-valued signals has information for the plaintext (data) and the communication basis, simultaneously. The information of the basis corresponds to that of the running key. Therefore, if the each signal value can be determined by 2M-ary detection, the plaintext and running key sequence information can be obtained directly.

In the above scheme, the information of binary plaintext in neighboring signals is completely masked by noises, but the information of the running key of *M* values has finite ambiguity, because the Γ is small. In such a situation, it is easy for Eve to memorize the digital ciphertext sequence of the 2M-ary signals containing errors. Eve can try KPA based on exhaustive search for all binary decisions similar to Bob on the stored sequence. This is called a legitimate receiver simulation attack.

Here, let us denote a rough approximate analysis for intuitive understanding. If the length of known plaintext is |X|=|K|/logM which corresponds to the equivalent key length and also equals the length of the ciphertext, then the probability of the correct decision for the running key $K_{R}=\left\{\alpha e^{i\theta_{m}}\right\}$ of |K|/logM length is
(49)$$P\left(K_{R}|Y^{E_{q}}\right)≅{\left(\Gamma /2\right)}^{-\frac{\left|K\right|}{\log M}}\ll 1$$
Equation (Equation 49) means that the following number of keys corresponds to the correct plaintext of |K|/logM length in the exhaustive search.
(50)$${\left(\Gamma /2\right)}^{\frac{\left|K\right|}{\log M}}$$
This means
(51)$$n_{1}^{Q}>\frac{\left|K\right|}{\log M}$$
In other words, these features correspond to the property of the random cipher based on the degeneracy structure [10]. On the other hand, if there is no error, the cipher is
(52)$$\begin{array}{ccc} & & P(K_{R}|Y^{E_{q}})=1,\\ & & n_{1}^{Q}=\frac{\left|K\right|}{\log M} \end{array}$$
The above characteristics indicate that the criticism stating that the standard quantum stream is the same as the conventional cipher is incorrect.

#### 7.1.2. Fast Correlation Attack

Let us assume that no restriction is placed on the known plaintext length. In this situation, one can consider the running key sequence (LFSR output) as a linear code and attempt a fast correlation attack on the received sequence with errors. The 2M-valued running key sequence of the LFSR output can be stored with errors. Then the 2M-valued sequence is converted to binary values and a fast correlation attack is performed. The performance is evaluated by the unicity distance. From the definition of unicity distance, we can obtain the channel capacity of Eve’s measurement channel. In general, the standard quantum stream cipher does not have sufficient performance against the fast correlation attack when Γ is small [28]. The concrete example to overcome this will be shown in the subsequent section.

#### 7.1.3. Secret Key Leakage Attack

Assume that Eve memorizes the digital ciphertext sequence of 2M-ary signals containing errors. After communication, we assume that Eve can obtain the secret key of PRNG. Eve will try an attack to estimate the plaintext that uses the correct secret key to the measured sequence with errors. That is, Eve can adopt the threshold for the binary decision depending on the correct running key, and she collects the plaintext sequence. The performance against such an attack is the most important in the generalized Shannon random cipher. Unfortunately, the standard quantum stream cipher does not have immunity against this type of attack.

### 7.2. Role of Additional Randomization Technique

In generalized random ciphers, the ciphertext or running key sequence obtained by an eavesdropper is perturbed by a true noise (such as quantum noise). If the noise effect is small, it naturally cannot provide sufficient information-theoretic security from the above theory. For realistic applications, it is necessary to develop techniques to reduce the channel capacity in Equation (Equation 31). Since noise must physically appear, such a generalized random cipher can be realized only at the physical layer.

The problem to make increasing noise, as described above, has little precedent in information theory and is the exact opposite of what has been done so far. That is, the technological development is to increase the noise effect of the eavesdropper, but it does not affect the legitimate communicator. In the channel model of the receiving process of the eavesdropper, the research to reduce its maximum mutual information is called “Randomization technology”, and several examples have already been studied. Specific models will be presented later, but further research is expected.

## 8. Randomizations Towards Generalized Quantum Stream Ciphers

The standard quantum stream cipher introduced in the previous section is highly practical and has affinity with real communication networks. However, it may have weaknesses in terms of the quantitative security evaluation. Randomization techniques are needed to realize a generalized quantum stream cipher with high performance. This section presents some examples of randomization methods and gives a rough description of the improvements achieved by each method.

### 8.1. Overlap Selection Keying (OSK)

The OSK is a mechanism to randomize the geometrical relation between the data (plaintext) and signal value of the communication basis, patented by Tamagawa University (Patent number 4451085, 27 June 2003) [29]. It is to randomize the relation between the data and given basis based on a sequence from a branch of the PRNG. By this method, the correlation of the geometrical relation of the data and communication basis is broken inside of the region of quantum noise masking Γ. So, we have the following performance instead of Equation (Equation 49):(53)$$P\left(K_{R}|Y^{E_{q}}\right)≅{\left(\Gamma \right)}^{-\frac{\left|K\right|}{\log M}}$$
In addition, the KPA is converted to the ciphertext-only attack. So the legitimate receiver simulation attack does not work. Thus, the unicity distance for KPA is guaranteed to have the following performance.
(54)$$n_{1}^{Q}\gg \left|K\right|$$

The other effect of OSK is that plaintext is automatically encrypted with pseudo-random numbers, and the plaintext itself becomes a kind of ciphertext called Y-00 plaintext.

### 8.2. Deliberate Signal Randomization (DSR)

Assume that a communication basis consists of a binary phase shift signal. A basis is randomly selected by the running key sequence from PRNG, and one signal for data is transmitted by the selected basis. Then, *M* signals are located on the upper plain of the phase space, and the other *M* signals are located on the lower plain of the phase space [10]. Here, phase space means a space by quadrature amplitude $X_{C}$ and $X_{S}$.

Even if M≫1, the masking effect Γ by quantum noise in Eve’s receiver is not enough, because the quantum noise is small. To enhance Eve’s error, the signal for transmission is randomly shifted by true noise on the upper plain when the signal belongs to the upper plain, and is shifted on the lower plain when it belongs to the lower plain. This scheme is called Deliberate Signal Randomization (DSR) which corresponds to “private randomization”. The strength of DSR is described by $|R_{p}|$. As a result, the masking region is enhanced by $\sigma |R_{p}|$, where σ is the quantum noise effect. This was proposed by Yuen in 10 November 2003 [30]. When we adopt OSK and DSR with $\sigma |R_{p}|=M$, we may have the ideal performance such as
(55)$$P\left(K_{R}|Y^{E_{q}}\right)≅2^{-|K|}$$
Thus, this may improve the unicity distance in Equation (Equation 54). The concrete example will be given in the subsequent section.

### 8.3. Quantum Noise Diffusion Mapping

The unicity distance of the generalized random cipher may be evaluated by the channel capacity of Eve from Equation (Equation 31). In this theory, it can be regarded that the linear code has the initial value of LFSR as information propagates through the noisy channel. Thus, the decoding capability, as the decryptability, at that time is evaluated by the channel capacity. In other words, as the length of the LFSR increases and the rate decreases, it becomes smaller than the channel capacity and the decoding accuracy increases. To obtain a large unicity distance, it is necessary to reduce the eavesdropper’s channel capacity by additional randomization like DSR. However, there is a trade-off in that it requires sacrificing the communication performance of the legitimate communicator. Therefore, a method to increase the unicity distance while keeping the amount of noise fixed is the subject of research.

In 2007, Hirota–Kurosawa proposed a method to introduce a mapping mechanism from a sequence of PRNG to actual quantum states, such that the input to the noisy channel is regarded as a nonlinear code [31]. This is achieved by re-diffusing the signal masked by a small quantum noise. This mechanism renders the fast correlation attack inoperative. In addition, it provides the immunity against algebra attacks. The detailed information-theoretic analysis of this method is still incomplete, but we look forward to further research.

### 8.4. Phase Masking by Symplectic Matrix

A theory for the randomization of the code form of coherent states is given by Sohma [32]. It consists of the unitary operator U associated with a symplectic transformation in which any unitary operator composed of beamsplitters and phase shifters can be described by a symplectic transformation. First, let us consider the general code form of the coherent state, as follows:(56)$$\left|\phi >=\right|\alpha_{1}>\left|\alpha_{2}>\ldots \right|\alpha_{N}>$$
From the Stone–von Neumann theorem [33], the quantum characteristic function for the class of quantum Gaussian state is given as follows:(57)$$\Phi \left(z\right)=TrU|\phi ><\phi |U^{†}V\left(z\right)=Tr\left|\phi ><\phi \right|V\left(L^{T}z\right)$$
where
(58)$$\begin{array}{ccc} V(z) & = & \exp\{iR^{T}z\} \end{array}$$
(59)$$\begin{array}{ccc} R & = & {\left[\left(q_{1},p_{1}\right),\ldots ,\left(q_{N},p_{N}\right),\right]}^{T} \end{array}$$
and where $\left(q_{i},p_{i}\right)$ are the canonical conjugate operators. Then L is a symplectic matrix, and it is given by
(60)$$\begin{array}{ccc} & & L=\left(\begin{array}{cccc} r_{11}e^{i\theta_{11}} & \ldots & \ldots & r_{1N}e^{i\theta_{1N}}\\ r_{21}e^{i\theta_{21}} & \ldots & \ldots & r_{2N}e^{i\theta_{2N}}\\ \vdots & \ldots & \ldots & \vdots \\ r_{N1}e^{i\theta_{N1}} & \ldots & \ldots & r_{NN}e^{i\theta_{NN}} \end{array}\right) \end{array}$$
Here, let us denote a vector of complex amplitudes α, as follows:(61)$${\vec{\alpha }}_{in}=\left(\alpha_{1},\alpha_{2},\ldots ,\alpha_{N}\right)$$
then, we have the following relation.
(62)$${\vec{\alpha }}_{out}=L{\vec{\alpha }}_{in}=\left(\alpha_{1}^{out},\alpha_{2}^{out},\ldots ,\alpha_{N}^{out}\right)$$
As a result, the unitary transformation for the coherent state sequence is given as follows:(63)$$U|\phi >=|\phi_{out}>=|\alpha_{1}^{out}>\left|\alpha_{2}^{out}>\ldots \right|\alpha_{N}^{out}>$$
Thus, by scrambling the elements of Equation (Equation 60) with pseudo-random numbers, one can construct codes with any waveform. This technology is useful to realize the coherent PPM scheme proposed by Yuen [30].

## 9. Generalized Quantum Stream Cipher of Type-I and Its Performance

A quantum stream cipher that adds randomization techniques like OSK and DSR to the standard quantum stream is called a Type-I generalized quantum stream cipher. In this section, we show the scheme and its performance.

### 9.1. Communication Scheme

Let us describe the communication scheme for phase shift keying (PSK). A running key sequence is generated by PRNG with a short secret key. The selection scheme of communication basis by the running key is the same as the standard quantum stream cipher. In addition, several randomizations described above may be installed. Then, 2M-valued signals are transmitted. Bob can adopt the binary quantum optimum receiver, but in practice he can adopt an optical heterodyne receiver.

In the latter case, the output of the receiver is an analog electrical current consisting of signal and quantum noise. For decoding, the threshold for binary decisions is controlled based on the running key. Using the selected threshold, the binary decision is performed to obtain data. This type of communication scheme can be realized by the current optical communication technology.

### 9.2. Unicity Distance of Known Plaintext Attack

Here, let us assume a phase shift keying (PSK) quantum stream cipher with DSR and OSK. Eve is forced to adopt a detection of 2M-ary signals and proceeds with the conventional fast correlation attack. Once the eavesdropper’s channel is set, the lower bound of the unicity distance is obtained by its channel capacity. The optimum condition of maximum mutual information for multi-valued quantum state signals is given by Holevo [17], and the optimum POVM is given by Osaki [34], based on the prediction of Fuchs–Peres [35], as follows:

**Theorem 7.** 
*{Osaki}: The optimum POVM for maximum mutual information is given by the quantum minimax detection operator when the signal set is covariant and linearly independent.*


The numerical performance has been verified for the maximum mutual information property based on the above theorem. As a result, when M≫1, it is possible to approximate its properties by the heterodyne receiver. In fact, M≫1 means that the signal set is regarded as an almost analog signal. To confirm it, we can estimate the optimality for an analog signal by the following theorem [36].

**Theorem 8.** 
*{Yuen·Lax}: The estimation bounds for complex amplitudes are given by the following formula.*

(64)
$$Var\left(\hat{\alpha }\right)\geq \frac{1}{Tr\rho LL^{†}}$$

*where the right logarithm derivative is defined by*

(65)
$$\frac{\partial \rho }{\partial \alpha }=L^{†}\rho$$

*And its solution is as follows:*

(66)
L=a

*where a is a photon annihilation operator, and it corresponds to a heterodyne measurement.*


So, we can assume that the eavesdropper adopts the heterodyne receiver, which is the highest performance for asynchronous quantum state signals. In this case, the eavesdropper receives 2M original phase signals to obtain *M*-valued information on the running key. The signal distance between signals is $\pi \sqrt{S}/M$, where S=|α| is the signal strength, and σ is the masking effect of the signal by quantum noise. The amount of signal masking is $\Gamma_{g}=2M\sigma /\pi \sqrt{S}$. When the strength of DSR that spreads the quantum noise effect is $|R_{p}|$, we have $\Gamma_{g}=2\left|R_{p}\right|\sigma M/\pi \sqrt{S}$. Here, the range of DSR can be set as follows.
(67)$$1\leq \sigma |R_{p}|<\frac{1}{2}\pi \sqrt{S}$$
The equivalent quantum noise of the optical heterodyne measurement is σ=1, and the maximum mutual information in the wedge approximation is [10]
(68)$$C_{Hetero}≅\log_{2}\frac{\pi \sqrt{S}}{2|R_{p}|}$$
(the exact communication channel capacity will be reported separately). Then the generalized unicity distance for KPA is
(69)$$n_{1}^{Q}>\frac{\left|K\right|}{\log_{2}\frac{\pi \sqrt{S}}{2|R_{p}|}}$$
This is called the Nair–Yuen formula [10]. If the strength of DSR is $|R_{p}|\;>\;\frac{1}{4}\pi \sqrt{S}$, the generalized unicity distance is
(70)$$\left|K\right|\ll n_{1}^{Q}\leq 2^{\left|K\right|}$$
This cannot be achieved using only a mathematical cipher. Thus, this scheme has advantages over conventional cryptographic mechanisms in terms of information-theoretic security. Although it is not the ultimate one, this type of system is expected to play a significant role in assuring the security of real optical communication systems.

### 9.3. Coherent Pulse Position Modulation Method

There is a method to realize the Type-I quantum stream cipher without using the above randomization technique. It is called the coherent pulse position modulation (CPPM). This mechanism takes information as *M*-valued and configures it as a transmitted signal system with *M*-ary PPM. The *M*-ary PPM signals are then spread like a pseudo-waveform signal into the *M*-ary slot by unitary transformations driven by a pseudo-random number generator. This scheme asymptotically approaches an eavesdropper’s ciphertext error of 1 as *M* increases. At the same time, however, it has the disadvantage of infinite baseband bandwidth, which makes it impractical. Recently, we have developed a method to avoid the bandwidth explosion. Details will be presented in another paper.

## 10. Generalized Quantum Stream Cipher of Type-II and Its Performance

In this section, we discuss a higher-security performance scheme than the Type-I system. When a secret key for the cipher is stolen after communications, the conventional cipher can be decrypted correctly. The most important feature of the generalized quantum stream cipher is that the cipher may not be decrypted even when the secret key is stolen. We discuss in this section “a conceptual model” to show that there exists secure communication even if the secret key is stolen.

### 10.1. Communication Scheme

Here we show that there exists a scheme that is resistant to the secret key leakage attack. Let us assume that a channel between Alice and Bob is low-loss.
(a)Alice uses PSK (phase shift keying). The set of the communication basis consists of two coherent states with small angles, as follows (See Figure 9):
(71)$$\begin{array}{ccc} Ba(1) & = & \{|\alpha_{1}>,|\alpha_{1}e^{i\Delta \theta }>\},\\ Ba(2) & = & \{|\alpha_{2}e^{i2\Delta \theta }>,|\alpha_{2}e^{i3\Delta \theta }>\},\\ Ba(3) & = & \{|\alpha_{3}e^{i4\Delta \theta }>,|\alpha_{3}e^{i5\Delta \theta }>\},\\ & & \vdots \\ Ba(M) & = & \left\{|\alpha_{M}e^{i(M-1)\Delta \theta }>,|\alpha_{M}e^{iM\Delta \theta }>\right\} \end{array}$$
where Δθ=π/M. When $\alpha =\alpha_{m},\forall m$, a set of 2M signals becomes covariant by Equation (Equation 40). This communication basis is selected by a running key sequence. Then, plaintext is set to the selected basis, the same as in Type-I, or the quantum cipher text is generated by the unitary transform $U\left(x\right)U\left(k^{R}\right)$ to coherent states controlled by the running key sequence.(b)Bob has the same unitary transform $U(k^{R})$, and it inversely transforms the input quantum states depending on the running key sequence. The output quantum states from the inverse unitary transformation can be regarded as binary quantum states.(c)Bob adopts the quantum optimum receiver with the Helstrom limit for these binary quantum states. The concrete system is called the Dolinar receiver. The average error for the data x=0,1 is independent of the basis and it is given by Equation (Equation 44), as follows:
(72)$${\bar{P}}_{e}^{B}=\frac{1}{2}\left\{1-\sqrt{1-\left|<\alpha \right|-\alpha e^{i\Delta \theta }{>|}^{2}}\right\}$$The amplitude and phase difference are designed so that the above equation holds sufficiently small.(d)Eve has to discriminate 2M-valued coherent state signals. Since Eve does not know the a priori probability distribution for the 2M-valued signals, she has to adopt a quantum minimax rule, as follows:
(73)$${\bar{P}}_{e}^{E}=\max_{\left\{\xi \right\}}\min_{\left\{\Pi \right\}}\left[1-\sum_{m=1}^{2M}\xi_{m}Tr{\tilde{\rho }}_{m}^{E}\Pi_{m}\right]$$
where $\left\{{\tilde{\rho }}_{m}^{E}\right\}$ is a set of Equation (Equation 71).

**Figure 9 entropy-26-00983-f009:**
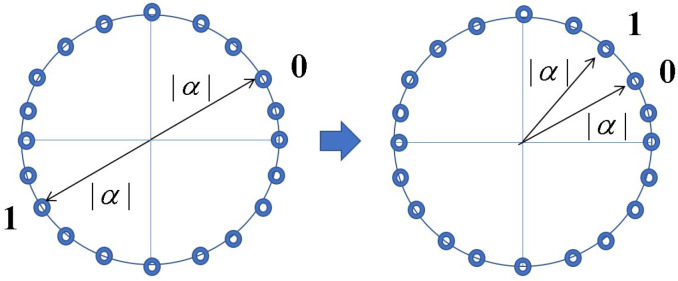
The signal arrangement above is the conventional form in the PSK scheme, while the one below is the proposed form; 0 and 1 are the information of the plaintext.

### 10.2. Secret Key Leakage Attack

Here, we show that the system is resistant to the secret key leakage attack. At first, Eve has to store a sequence of 2M-valued signals received by the above quantum minimax receiver to try the crypto analysis. Once Eve has the secret key and the correct running key sequence after the measurement, she can try a classical binary decision scheme on sequences with errors in the 2*M* values for each slot.

Let us show how it works. When a set of quantum states is a covariant, the solution of the minimax rule is equivalent to the quantum Bayes rule with the worst a priori probability distribution: $\xi_{m}=1/2M,\forall m$ [26]. When M≫1, one can approximate the error performance based on quantum noise by the analog version in the quantum decision theory. The solution becomes the heterodyne receiver. Thus, let us assume that Eve adopts the heterodyne receiver with an analog-to-digital converter with infinite bandwidth. She can store the almost analog signal consisting of signals and quantum Gaussian noise.

When Eve obtains the secret key and the correct running key sequence, she adopts the binary threshold decision based on the classical Bayes rule depending on the running key. It corresponds to a model for the binary decision for binary signals with Gaussian quantum noise, because the uncertainty caused by pseudo-random numbers will disappear. The error probability for the binary data (plaintext) is
(74)$${\bar{P}}_{e}^{E}\left(x\right)≅\mathrm{erfc}\left\{\frac{\left|\alpha |(1-\cos\Delta \theta \right)}{\sigma_{eq}}\right\}$$
The relation between Bob’s error, Equation (Equation 72), and Eve’s errors, Equation (Equation 74), with secret the key provided after communication for the plaintext (data) is as follows:(75)$${\bar{P}}_{e}^{E}\left(x\right)\gg {\bar{P}}_{e}^{B}\left(x\right)$$
Thus, the sequences of plaintexts that Bob and Eve can obtain become as follows:(76)$$\begin{array}{ccc} & & X_{n}^{B}≅x_{1},x_{2},x_{3},\ldots ,x_{n} \end{array}$$(77)$$\begin{array}{ccc} & & X_{n}^{Eq}=x_{1}^{E_{q}},x_{2}^{E_{q}},x_{3}^{E_{q}},\ldots ,x_{n}^{E_{q}}=x_{1}⊕q_{1},x_{2}⊕q_{2},x_{3}⊕q_{3},\ldots ,x_{n}⊕q_{n} \end{array}$$
where $x_{i}:\left\{0,1\right\}$, $q_{i}:\left\{0,1\right\}$. As a result, Eve’s sequence consists of the plaintext *X* and the quantum error sequence.

Here, we can regard the quantum error as the random key: $K_{q}=q_{1},q_{2},\ldots ,q_{n}$. The structure of Eve’s sequence of plaintext, Equation (Equation 77), is equivalent to the Shannon cipher with the key sequence $K_{q}$. So the unicity distance of the final sequence for Eve is as follows.
(78)$$n_{2}^{Q}=\frac{H(K_{q})}{1-H(p)}$$
where $H(K_{q})$ is the entropy of the quantum error sequence and H(p) is the entropy per symbol of the plaintext.

Thus, in principle, there exists the generalized Shannon random cipher with the following performance:(79)$$\begin{array}{ccc} H(X|Y^{B},K) & ≅ & 0,\;for\;\mathrm{Bob} \end{array}$$(80)$$\begin{array}{ccc} H(X|Y^{E},K) & > & 0,\;for\;\mathrm{Eve} \end{array}$$

However, we emphasize that this is intended to provide theoretical evidence and is applicable only to channels with very low losses. Therefore, more careful research is needed for practical application.

## 11. Generalized Quantum Stream Cipher of Type-III and Its Performance

The principle of the previous schemes was to adopt a pseudo-random number generator (PRNG) for the diffusion of the quantum noise effect. Here, we show new schemes such that the quantum ciphertext is constructed by directly mapping the plaintext regularly to the quantum state signal without an encryption mechanism by the PRNG and secret key.

### 11.1. Communication Scheme as Protocol 1

First, we consider the encryption mechanism such that the advantage creation is given only by the difference of Bob’s and Eve’s receiving mechanisms. Let us assume that Alice can control the a priori probability of *M*-valued coherent state signals. So, Bob can use the quantum Bayes rule because Bob knows the a priori probability of the ciphertext, and Eve is forced to use the quantum minimax rule because she does not know the a priori probability of the ciphertext. This corresponds to the advantage creation based on only error performances between the quantum Bayes rule and quantum minimax rule. The communication scheme of the protocol is as follows:(a)The plaintext consists of a combination of *J* bits $\left(x_{1},x_{2},x_{3},\ldots ,x_{J}\right),x_{j}=\left\{0,1\right\}$. The information data becomes $M=2^{J}$, as follows:
(81)$$\begin{array}{ccc} X_{1} & = & \left(1,0,0,\ldots ,0\right)\\ X_{2} & = & \left(0,1,0,\ldots ,0\right)\\ & & \vdots \\ X_{M} & = & \left(1,1,1,\ldots ,1\right) \end{array}$$Next, assume that Alice can control the prior probability distribution $\left\{\xi_{m}^{A}\right\}$ of *M*-valued coherent state signals. However, the structure of the quantum state set $\left\{\rho_{m}^{A}\right\}$ will be exposed. In other words, the only thing unknown to Eve is the a priori probability distribution.(b)The *M*-valued plaintext is mapped regularly to one of the *M*-valued quantum states as follows:
(82)$$\rho_{m}^{A}=\left|\alpha_{m}><\alpha_{m}\right|,\;m=1,2,3,\ldots ,M$$For example, an *M*-ary PSK scheme can be used, but the signal constellation may not be covariant.(c)Bob has information for the a priori probability for *M* signals and has information about the structure of the quantum state ensemble $\rho_{m}^{A}$. Then, he can adopt the quantum Bayes decision rule. His error probability is given by
(83)$${\bar{P}}_{e,Bayes}^{B}=\min_{\left\{\Pi \right\}}\left[1-\sum_{m=1}^{M}\xi_{m}^{A}Tr\rho_{m}^{A}\Pi_{m}\right]$$Eve knows the structure of quantum state signals, but she does not know the a priori probability distribution. So, she has to adopt the quantum minimax decision rule as follows:
(84)$${\bar{P}}_{e,min}^{E}=\max_{\left\{\xi \right\}}\min_{\left\{\Pi \right\}}\left[1-\sum_{m=1}^{M}\xi_{m}Tr\rho_{m}\Pi_{m}\right]$$Then, her error probability is given by the quantum Bayes rule with the worst a priori probability. We wish to enlarge the absolute quantity of Eve’s error. To achieve this, we can adopt the concept of covariant and non-covariant signals [37,38,39]. For discrete signals, the criterion of these is given by the following [40].

**Theorem 9.** 
*{Usuda·Takumi}:*
*M-ary signals $\left\{|\psi_{m}>,m=1,2,3,\ldots ,M\right\}$ are group covariant with respect to a group (G;∘) of order M if and only if the following relation holds:*(85)$$<\psi_{k\circ m}\left|\psi_{k\circ l}>=<\psi_{m}\right|\psi_{l}>,\forall k,l,m\in G$$*where G={1,2,3,…M} and *∘ *is the operation of the group G.*

The minimax decision rule gives a solution assuming an a priori probability distribution that gives the maximum value of the Bayes rule solution. Here, when the signal system is not group covariant, the worst a priori probability distribution of minimax is not uniform [26]. According to the general theory of quantum detection theory [17,19], we have the following relation.
(86)$${\bar{P}}_{e,Bayes}^{E}\left(Covariant\right)={\bar{P}}_{e,minimax}^{E}\left(Covariant\right)<{\bar{P}}_{e,minimax}^{E}\left(Noncovariant\right)$$
That is, when the set of quantum states is not covariant, the error performances for the minimax rules themselves is greatly degraded, depending on the signal properties [26,41]. Thus, we can have the following relation.
(87)$${\bar{P}}_{e,Bayes}^{B}\left(Noncovariant\right)<{\bar{P}}_{e,minimax}^{E}\left(Noncovariant\right)$$

Using this property, the system uses a signal set that is not group covariant and transmits with an a priori probability distribution $\xi_{m}^{A}$ that maximizes the characteristic difference between Bayes and minimax. The optimization problem is given as follows:(88)$$\max_{\left\{\rho_{m}\right\}}\max_{\left\{\xi_{m}^{A}\right\}}|{\bar{P}}_{e,Bayes}^{B}\left(Noncovariant\right)-{\bar{P}}_{e,minimax}^{E}\left(Noncovariant\right)|$$
where $\xi_{m}^{A}$ is the a priori probability distribution that Alice can control, and the relation among the a priori probability distributions is $\xi_{m}^{A}=\xi_{m}^{B}\neq \xi_{m}^{E}$. $\left\{\rho_{m}\right\}$ is a structure of non-covariant states. Here, the worst a priori probability distribution $\xi_{m}^{E}$ for the minimax is determined only by the structure of non-covariant quantum states. Thus, even if the legitimate communicators do not have PRNG with the secret symmetric key, we can realize an encryption scheme based on the advantage creation due to the Holevo–Yuen theory. The details of this property require numerical analysis and it will be shown in the subsequent paper.

### 11.2. Communication Scheme as Protocol 2

Here, we consider the second candidate of the encryption without PRNG applicable to space communication. In general, space communication is modeled by a continuous waveform communication model [42]. In such purposes, a wire-tap channel model assumes poor signal-to-noise ratio conditions for eavesdroppers under natural conditions. However, the wire-tap channel models based on the Holevo–Yuen theory are different to conventional theories, because they are characterized by creating an advantage for legitimate communicators by legitimate communicators. Let us discuss the basic concept of the wire-tap channel scheme in the sense of the advantage creation principle based on the quantum communication theory by Holevo–Yuen.

The conventional wire-tap channel model requires a special channel such that the signal-to-noise ratio of Bob is greater than that of Eve, as the system requirement. We replace its condition with the notion of advantage creation in the quantum measurements. According to the quantum Shannon theory established by Holevo and others [16], the capacity formula for the lossy Gaussian noise channel for coherent states is given as follows [43]:

**Theorem 10.** 

{Holevo·Sohma·Hirota}


*The capacity formula of the quantum lossy Gaussian noise channel for coherent state signals is given as follows:*

(89)
$$C_{H}=\log\left(1+\frac{S}{1+<n>}\right)+S\log\left(1+\frac{1}{S+<n>}\right)-<n>\log\left(\frac{1+\frac{S}{<n>}}{1+\frac{S}{1+<n>}}\right)$$

*where S and <n> are average photon numbers of the received signal and additive noise, respectively.*


The above formula is in general greater than the Shannon classical capacity. To achieve this Holevo capacity in the realization stage, the communicators must have prior knowledge of the quantum signal structure, time–phase synchronization and other various conditions. The capacity can only be achieved by adopting a quantum optimum measurement under those conditions. The time–phase synchronization between Alice and Bob is available, but Eve cannot obtain such system parameter information.

The secret capacity is defined as follows:(90)$$C_{S}=\max_{\left\{\xi \right\},\left\{\Pi^{B}\right\}}I^{B}\left(X,Y\right)-\max_{\left\{\xi \right\},\left\{\Pi^{E}\right\}}I^{E}\left(X,Y\right)$$
where $\left\{\Pi^{B}\right\}$ and $\left\{\Pi^{E}\right\}$ are POVMs for Bob and Eve, respectively. The mutual information for Eve is formulated under the condition that Eve does not know the system parameter information. If the system parameter is not known, a heterodyne receiver would be optimal. Thus, Eve’s POVM is restricted to the heterodyne. With this differentiation, it is possible to construct a modified wire-tap channel communication scheme.

A conjecture of the concrete formula of the secret capacity for this model based on the coherent state is as follows [44].
(91)$$\begin{array}{ccc} & & C_{S}=C_{H}-C_{Shannon}=\log\left(1+\frac{S^{B}}{1+<n{>}^{B}}\right)+S^{B}\log\left(1+\frac{1}{S^{B}+<n{>}^{B}}\right)\\ & & -<n{>}^{B}\log\left(\frac{1+\frac{S^{B}}{<n{>}^{B}}}{1+\frac{S^{B}}{1+<n{>}^{B}}}\right)-\log\left(1+\frac{S^{E}}{1+<n{>}^{E}}\right) \end{array}$$
where $S^{B}$ and $<n{>}^{B}$ are the average photon numbers of signal and noise for Bob. $S^{E}$ and $<n{>}^{E}$ are those for Eve. When the above formula is positive, one can implement the secure communication system in principle.

For the practical discussions, we need a discretization technique like the realization of classical waveform communication [42]. Then we need to construct a coding theory that creates an advantage for the legitimate communicators. It is equivalent to constructing the super-additivity of mutual information. The first challenge to clarify the effect of coding was made in [45].

## 12. Conclusions

In this paper, we explained that generalized quantum stream ciphers, which can improve the shortcomings of one-time pad ciphers, are realizable by applying quantum effects. The most important claim of this paper is that the quantum stream cipher based on the Holevo–Yuen theory is guaranteed to be information-theoretically secure, even though it consists of a PRNG with a short secret key. Figure 10 shows the difference in principles that guarantee the security. To demonstrate the performance of such ciphers, we have introduced the generalized unicity distance and showed some examples. A more detailed theoretical analysis needs to be developed for the realization of the ultimate performance, such as for Type-II and Type-III.

On the other hand, a number of experimental studies for the standard quantum stream cipher have already been initiated on the basis of the above theory. As a result, promising results for practical application have been obtained by groups in the USA, Japan, and China [46,47,48,49,50,51,52,53]. In addition, the generalized quantum stream cipher with randomizations of Type-I has been implemented by the Futami group [54]. This is the first demonstration of a quantum stream cipher with sufficient information-theoretic security.

The purpose and methods of development of such research and development are quite different from those of quantum technology in physics, which currently has the largest research population. As a result, the concept of security and the method of proof are completely different, making some aspects difficult to explain. Figure 11 illustrates the difference in concepts. Figure 12 shows the summary of the comparisons among several schemes of stream ciphers.

Finally, another possibility of the generalized random ciphers is Shapiro’s scheme, the so-called quantum low probability of intercept [55], and Lloyd’s scheme [56]. These will be introduced in part III.

## Figures and Tables

**Figure 1 entropy-26-00983-f001:**
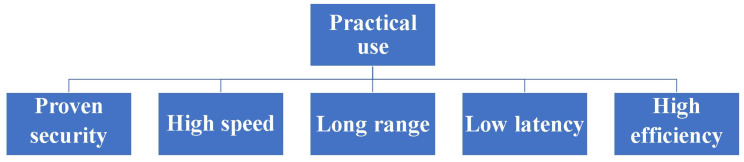
Requirements for cryptography to be used in the real world.

**Figure 2 entropy-26-00983-f002:**
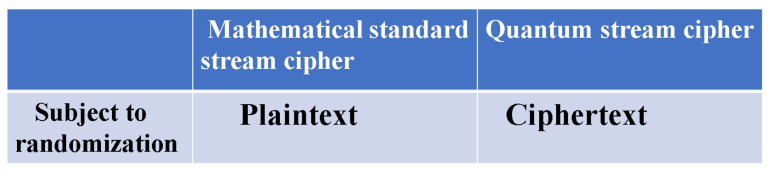
Difference in randomization for mathematical cipher and quantum stream cipher.

**Figure 3 entropy-26-00983-f003:**
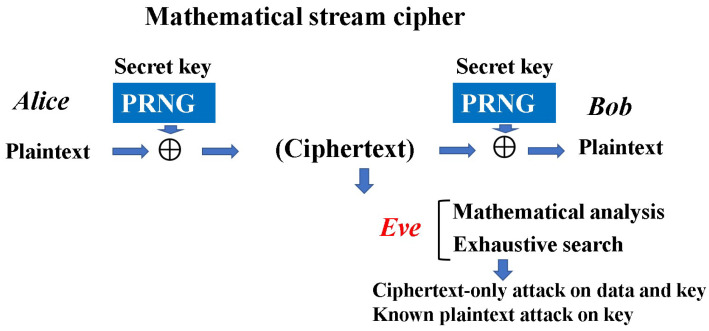
Structure of mathematical stream cipher.

**Figure 4 entropy-26-00983-f004:**
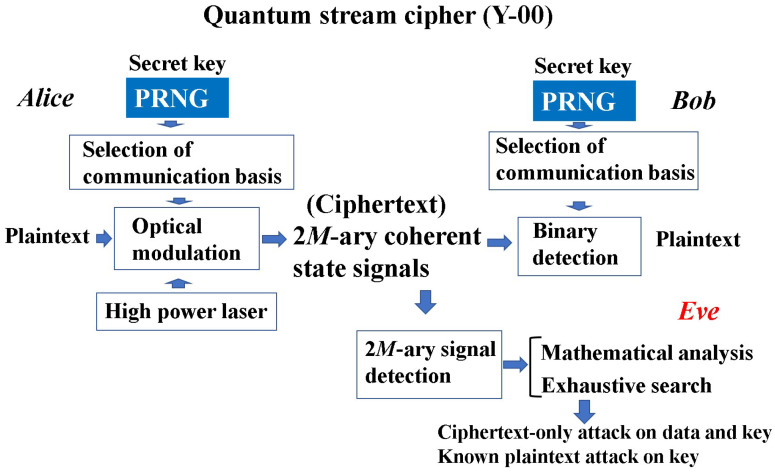
Structure of the standard quantum stream cipher, so-called the Y-00 protocol. The light source is an ordinary high-powered laser. The security is evaluated by Eve’s error when she observes the ciphertext. If the error is zero, it is equivalent to the security of PRNG.

**Figure 5 entropy-26-00983-f005:**
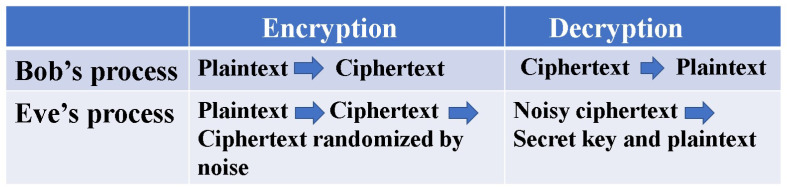
Encryption and decryption process in quantum stream cipher for Bob and Eve. The ciphertext is generated at the transmitter by an optical *M*-ary modulator for the laser light controlled by PRNG. It corresponds to the 2M-ary coherent state signal. The difference between Bob and Eve is whether the receiver has the correct PRNG and secret key. The difference gives the difference of error performance when Bob and Eve observe the ciphertext signals.

**Figure 6 entropy-26-00983-f006:**
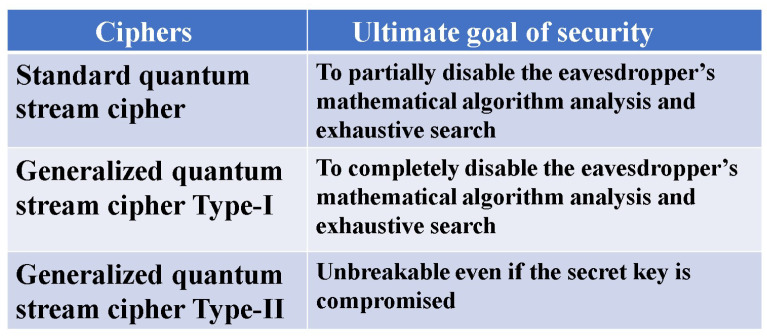
Targeted security of several quantum stream ciphers. The standard type is the scheme currently under development. Type-I causes Eve to receive a complete random ciphertext. Its realization is the latest goal of our experimental research.

**Figure 7 entropy-26-00983-f007:**
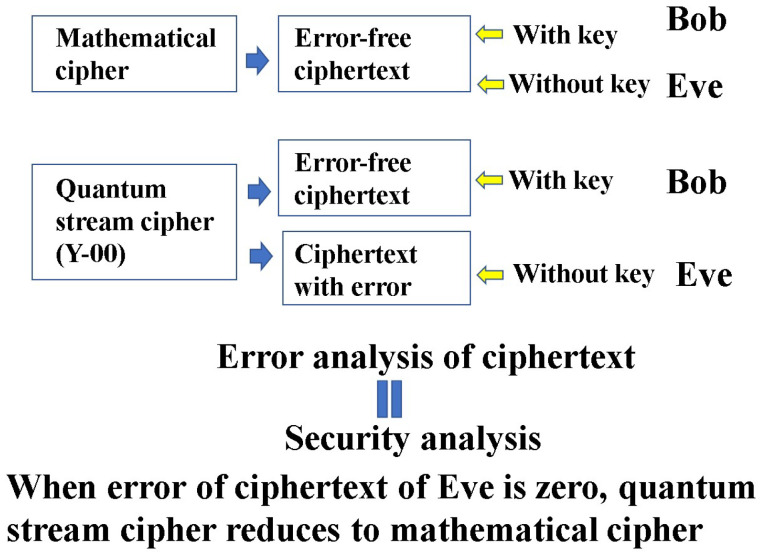
The principle of the security analysis. The security is governed by the error performance of the eavesdropper’s ciphertext based on quantum communication theory. The subsequent process of the security analysis is the same as that of the mathematical cipher under the error.

**Figure 8 entropy-26-00983-f008:**
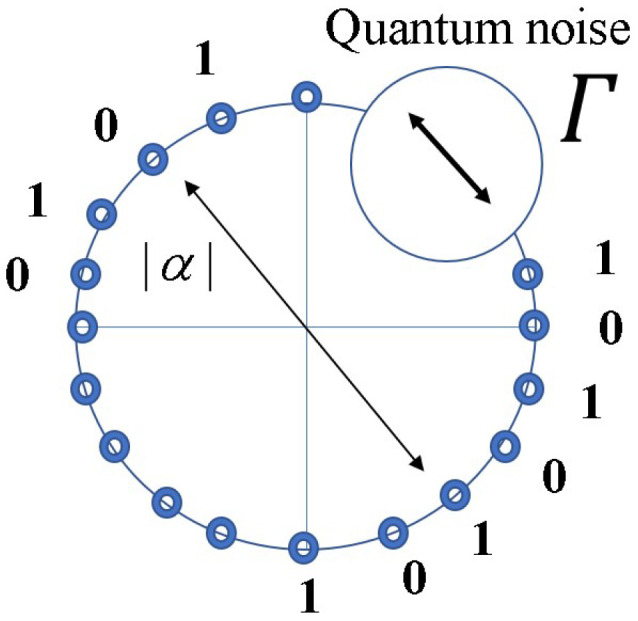
The signal arrangement in the PSK scheme. Points on the circle are signals in the phase space; 0, 1 are plaintext; and |α| is the amplitude of the laser.

**Figure 10 entropy-26-00983-f010:**
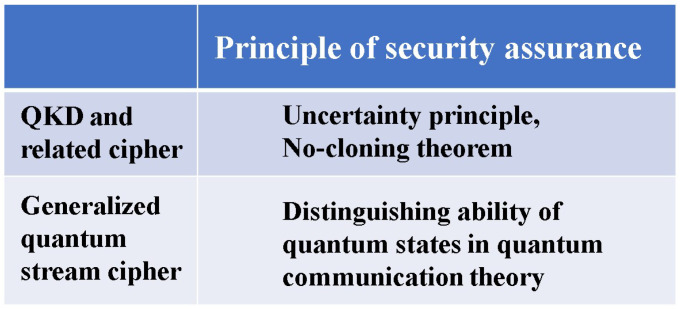
Essential difference in the principle of security assurance between schemes using microscopic and macroscopic quantum phenomena.

**Figure 11 entropy-26-00983-f011:**
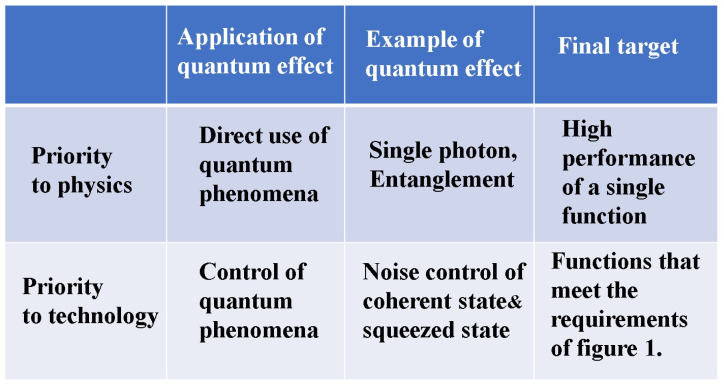
Essential differences in the realization method between schemes using microscopic and macroscopic quantum phenomena.

**Figure 12 entropy-26-00983-f012:**
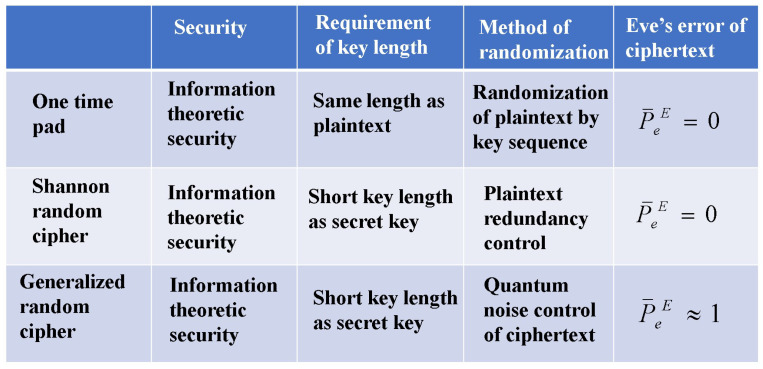
Essential differences in features for conventional mathematical cipher and quantum stream cipher to attain information theoretic security.

## Data Availability

Data are contained within the article.

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
