# Peer review of "Quantum Stream Cipher Based on Holevo–Yuen Theory: Part II"

_entropy, 2024, doi:10.3390/e26110983_

Round 1
Reviewer 1 Report
Comments and Suggestions for Authors
This paper collects a lot of information about randomized cryptography in one place, which seems to be a good reference for researchers in the field. They further describe a generalized quantum stream cipher that can maintain information theoretic security, at least in some limiting cases, even when the secret key is leaked. The paper seems to be a useful contribution and I can recommend publication. I have the following suggestions as places where the authors can consider making changes:
Page 2: -why can’t a mathematical cipher randomize ciphertext? For instance, a software implementation of a quantum stream cipher is possible (using a local noise source), and this could be regarded as a mathematical cipher (perhaps not according to their conception of what a mathematical cipher is, but a precise definition of what they mean by mathematical cipher may help here). Also, homophonic substitution is later defined as where the ciphertext is not uniquely determined by the key-plaintext pair-this seems like both a mathematical cipher and randomized ciphertext. Some more explanation of what the authors have in mind here may clarify their point.
Page 2: -“Applications of such a physical encryption technique appear to be limited, not general communication networks.” Can the authors explain what the limitations are in terms of network usage? Are there ways to work around these limitations? If not, how does this limit the range of applicability?
- it may help to precisely define known plaintext and ciphertext only attacks early on and at the same place in the paper. It seems the authors regard known plaintext as exactly known plaintext, and ciphertext-only as any kind of statistical knowledge of the plaintext (including zero knowledge). These are not the only definitions though, and sometimes a statistical knowledge of the plaintext is given its own terminology. However things are defined, it will help the reader to make it clear early on.
Author Response
Comment:
This paper collects a lot of information about randomized cryptography in one place, which seems to be a good reference for researchers in the field. They further describe a generalized quantum stream cipher that can maintain information theoretic security, at least in some limiting cases, even when the secret key is leaked. The paper seems to be a useful contribution and I can recommend publication. I have the following suggestions as places where the authors can consider making changes:
Page 2: -why can’t a mathematical cipher randomize ciphertext? For instance, a software implementation of a quantum stream cipher is possible (using a local noise source), and this could be regarded as a mathematical cipher (perhaps not according to their conception of what a mathematical cipher is, but a precise definition of what they mean by mathematical cipher may help here). Also, homophonic substitution is later defined as where the ciphertext is not uniquely determined by the key-plaintext pair-this seems like both a mathematical cipher and randomized ciphertext. Some more explanation of what the authors have in mind here may clarify their point.
Page 2: -“Applications of such a physical encryption technique appear to be limited, not general communication networks.” Can the authors explain what the limitations are in terms of network usage? Are there ways to work around these limitations? If not, how does this limit the range of applicability?
- it may help to precisely define known plaintext and ciphertext only attacks early on and at the same place in the paper. It seems the authors regard known plaintext as exactly known plaintext, and ciphertext-only as any kind of statistical knowledge of the plaintext (including zero knowledge). These are not the only definitions though, and sometimes a statistical knowledge of the plaintext is given its own terminology. However things are defined, it will help the reader to make it clear early on.
Reply:
Reply to Reviewer-1
Thank you very much for your valuable comments. Since the purpose of this paper is to integrate mathematical and physical ciphers, such comments are greatly appreciated.
A: Q-1
It is possible to randomize the ciphertext in a mathematical cipher. However, in that case, Bob and Eve also receive the same ciphertext. To bring out the characteristics of the cipher presented here, it is non-commensurable that Bob's ciphertext and Eve's ciphertext are different (Equation 18). Currently, this can only be achieved in a physical environment. We, too, are working on a way to achieve this in a software but have not yet succeeded. If it succeeds, this type of cipher will have a very wide range of applications and is a promising theme for the future.
A: Q-2
This cipher is not affected by the presence of optical amplifiers in the communication link, so it can be applied to any current optical network. However, it is difficult to apply it to wide-area networks because optical-electrical switching systems are currently the mainstream. It is necessary that optical-to-optical switching systems be developed in the future.
A: Q-3
I think this issue is important. We will consider it as a future issue. Since this paper deals with a lot of issues, it is difficult to make it a complete explanation, but we will take your comments into consideration and improve it further in the future.
Reviewer 2 Report
Comments and Suggestions for Authors
My comments are in the attached PDF.

Author Response
Comment:
In the manuscript titled: “Quantum Stream Cipher Based on Holevo-Yuen Theory
Part II,” the authors review the quantum stream cipher based on Holevo-Yuen
Theory.
Overall, after carefully reading this manuscript, I think the authors need to address
all of following comments before this article can be accepted for publication.
1. In section 5-line 383 of this paper, author wrote ”In order to guarantee ultrahigh
speed and long-distance transmission, it is necessary to adopt optical signals with
high energy, not single photon or entanglement light.”
In my viewpoint, this is not completely correct, as there are many high-performance
and long-distance quantum optical transmission via single photon or entanglement
light, such as in high-dimensional quantum key distribution (QKD) protocols: Phys.
Rev. A 87, 062322 (2013), New J. Phys. 17, 022002 (2015), Opt. Express 27, 17539
(2019), and Quan. Sci. Technol. 9, 015018 (2024); CV-QKD protocols: Phys. Rev.
Lett. 125, 010502 (2020), Sci. Adv. 10, eadi9474 (2024); MDI-QKD protocols: Phys.
Rev. Lett. 125, 260503 (2020), and Phys. Rev. Appl. 15, 064016 (2021). Hence, I
think authors should cite these corresponding papers in this section to better help the
readers to understand the current progress on the QKD protocols.
2. In section 6-line 446 of this paper, the authors introduce the foundation of
quantum communication theory for cryptanalysis.
However, I think the error correction codes are important as the foundation of
quantum communication and should be included in the beginning of this section. For
example, for the error correcting codes, high-dimensional QKD have also used the
low-density parity-check codes (LPDC) as error-correction codes, such as Efficient
information reconciliation for energy-time entanglement quantum key distribution,
In 2019 53rd Asilomar Conference on Signals, Systems, and Computers (pp. 1364
1368). IEEE, High-dimensional energy-time entanglement distribution via a
biphoton frequency comb, Conference on Lasers and Electro-Optics, OSA Technical
Digest (Optical Society of America, 2021), paper FF1A.7, and Time-Entanglement
QKD: Secret Key Rates and Information Reconciliation Coding, in IEEE
Transactions on Communications, 71, no. 12, pp. 7174-7188, Dec. 2023. Overall, I
suggest authors to include more information and references for the error correction
codes.
Reply:
Reply to Reviewer 2
The reviewer comments on the structure of this paper from the standpoint of a QKD expert. The cipher presented here is a symmetric key cipher that directly encrypts data, not a key transmission cipher. I would like to state first that QKD and the symmetric key cipher described in this paper follow completely different theories, so it is important to avoid direct comparison.
QKD is a very important subject, and we support its research. Its security guarantees are more demanding than those of symmetric key cipher and require a much more advanced theory than the one described here.
We plan to present the Holevo-Yuen theory of QKD in Part III. The reviewers' comments contain important details, and we will take them into account in our efforts to complete the paper.
A: Q-1
The ciphers presented here are intended to protect existing ultrahigh-speed, long-haul optical communication lines. That is, it is required to directly encrypt data at speeds of 10 Gbit/sec to 100 Gbit/sec in real time over distances of 1000 km on existing lines with at least optical amplifiers installed. We are developing QKD not for current line protection, but for the future quantum Internet, because there is a very strong trade-off between speed and distance. Stay tuned for Part III.
A: Q-2
The cryptographic schemes presented here are almost error-free for data between legitimate communicators because of the availability of strong lasers. Error-correcting codes are therefore not important. In QKD, they are very important because, besides their error-correcting function, they are also involved in guaranteeing security. In Part III, we will introduce the relationship between the two issues